# DNA Polymorphisms and mRNA Levels of Immune Biomarkers as Candidates for Inflammatory Postpartum Disorders Susceptibility in Italian Buffaloes

**DOI:** 10.3390/vetsci10090573

**Published:** 2023-09-14

**Authors:** Ahmed Ateya, Fatmah A. Safhi, Huda El-Emam, Muath Q. Al-Ghadi, Mohamed Abdo, Liana Fericean, Rada Olga, Ostan Mihaela, Manar M. Hizam, Maha Mamdouh, Eman M. Abu El-Naga, Walaa S. Raslan

**Affiliations:** 1Department of Development of Animal Wealth, Faculty of Veterinary Medicine, Mansoura University, Mansoura 35516, Egypt; hdemam@mans.edu.eg; 2Department of Biology, College of Science, Princess Nourah bint Abdulrahman University, P.O. Box 84428, Riyadh 11671, Saudi Arabia; faalsafhi@pnu.edu.sa; 3Department of Zoology, College of Science, King Saud University, P.O. Box 2455, Riyadh 11451, Saudi Arabia; malghadi@ksu.edu.sa; 4Department of Animal Histology and Anatomy, School of Veterinary Medicine, Badr University in Cairo (BUC), Cairo 11829, Egypt; mohamed.abdo@vet.usc.edu.eg; 5Department of Anatomy and Embryology, Faculty of Veterinary Medicine, University of Sadat City, Sadat City 32897, Egypt; 6Department of Biology and Plant Protection, Faculty of Agricultural Sciences, University of Life Sciences King Michael I, 300645 Timisoara, Romania; mihaelafericean@usab-tm.ro (L.F.); mihaela_ostan@usvt.ro (O.M.); 7College of Pharmacy, National University of Science and Technology, Nasiriyah 64001, Iraq; manar.m.belt@nust.edu.iq; 8Department of Physiology, Faculty of Veterinary Medicine, Benha University, Toukh 13736, Egypt; maha.mamdouh@fvtm.bu.edu.eg (M.M.); walaa.raslan@fvtm.bu.edu.eg (W.S.R.); 9Department of Theriogenology, Faculty of Veterinary Medicine, Aswan University, Aswan 81528, Egypt; e.abuelnaga@vet.aswu.edu.eg

**Keywords:** immune biomarkers, transcript levels, reproductive disorders, Italian buffaloes

## Abstract

**Simple Summary:**

Future reproduction methods for boosting crucial traits will likely be influenced by the proportion of genetic and phenotypic variances. Using PCR-DNA sequencing, the variations in the nucleotide sequences of the immunological genes between healthy buffaloes and buffaloes suffering from inflammatory reproductive disorders were discovered. When buffaloes experienced inflammatory reproductive disorders, they were more likely to express the examined genes. The molecular alterations may, therefore, provide crucial information about the relationships between the physiologies of the various reproductive pathways and help diagnose inflammatory postpartum disorders.

**Abstract:**

The immunological genes that may interact with inflammatory postpartum diseases in Italian buffaloes were examined in this study. A total number of 120 female Italian buffaloes (60 normal and 60 with inflammatory reproductive diseases) were employed. Each buffalo’s jugular vein was pierced to get five milliliters of blood. To obtain whole blood and extract DNA and RNA, the blood was placed within tubes containing sodium fluoride or EDTA anticoagulants. The immunological (*IKBKG*, *LGALS*, *IL1B*, *CCL2*, *RANTES*, *MASP2*, *HMGB1*, and *S-LZ*) genes’ nucleotide sequence differences between healthy buffaloes and buffaloes affected by inflammatory reproductive diseases were found by employing PCR-DNA sequencing. According to Fisher’s exact test (*p* ˂ 0.01), there were noticeably different probabilities of all major nucleotide changes spreading among buffalo groups with and without reproductive problems. Buffaloes were significantly more likely to express the examined genes when they had inflammatory reproductive diseases. The outcomes might support the significance of these markers’ nucleotide variations and gene expression patterns as indicators of the prevalence of inflammatory reproductive disorders and provide a workable buffalo management policy.

## 1. Introduction

The postpartum period, or the first 45 days following delivery, is the most stressful time due to decreased feed intake, endocrine and metabolic changes during parturition and lactation, and usually a multitude of physiological stressors with immunological roots accompanying it [1]. If these changes are not effectively treated, many inflammatory and non-inflammatory pathological illnesses will emerge during this time, including mastitis, endometritis, and hypocalcemia [2,3,4]. By lowering buffalo conception rates and ages and raising newborn mortalities, these illnesses have a deleterious impact on the flock breeding economic yield. Oxidative stress is the primary factor in immunosuppression and increases disease susceptibility during transition [5]. Due to the increased metabolic demands brought on by late pregnancy, delivery, and nursing beginning, reactive oxygen species (ROS) and lipid peroxidation are known to harm tissues, particularly immune cells [6]. The fight against oxidative stress brought on by free radicals involves preventive, repair, physical, and antioxidant defenses [7].

Buffaloes are predisposed to inflammatory reproductive disorders after delivery since the majority of adult buffalo problems frequently manifest within the first month [8,9,10]. Inflammation is important for repeating tissue receptiveness and founding conception in postpartum dairy cows [11], which is characterized by the exclusion of bacterial infections and renewal of the inflamed tissue. Due to a poor immunological response, cows may struggle to quickly clear infections [12]. Inflammatory reproductive problems are a result of a persistent inflammatory response in tissue [13], which is related to this. Dairy buffaloes are susceptible to bacterial reproductive infections during the time following parturition [14]. Since a variety of bacteria can be easily isolated from the reproductive lumen, it is thought that bacterial infections are the main cause of the majority of inflammatory reproductive ailments [15]. Inflammatory reproductive problems have also been connected to increased amounts of pro-inflammatory cytokine transcripts because of poor inflammation activation and bacterial clearance [16].

In mammals, there are two types of immunity: innate immunity and adaptive immunity. Both vertebrates and invertebrates have innate immunity, whereas only vertebrates have adaptive immunity [17]. Examples of various cascades that significantly affect the immune system include the nutritional stress of cattle production methods. It was previously believed that problems with general health at this time had a secondary impact on immunological response. However, when new cellular and molecular immunological pathways emerged, these issues and how they affected the immune system started to assume a more definite molecular reality [18]. Innate immunity supports host genetic resistance primarily, providing defense against pathogens without vaccination or exposure to illnesses [19]. Indeed, some genetic variants that may be responsible for variance in reproductive features have been highlighted by association studies [20]. We will probably be better able to recognize the genes and polymorphisms that are responsible for these traits as we are still researching how genes work and are expressed in relation to cow reproductive disorders. This could eventually result in more precise methods of genomic selection that are more effective across generations and breeds [21].

Given that all evolution results from genetic variants, describing the genetic makeup of a population necessitates an understanding of that group’s evolutionary history [22,23]. Additionally, the percentage of genetic and phenotypic differences is anticipated to have an impact on future reproduction techniques for enhancing critical features [24]. Because of this, the development in molecular genetics has given us the chance to find genes and their intersecting paths. The last is more insight into the physiological mechanisms behind these candidate genes and has the potential to be utilized as genetic indicators in MAS programs [25]. Remarkably, cattle populations have a significant number of significant genes. These important genes have been linked to various reproductive, illness, or production traits in the general population [26]. Additionally, animals with a high level of resistance expressed thousands of genes associated with immune function, according to gene expression studies [27,28]. Association analysis performed on a variety of animals typically identifies genetic variants that have an impact on phenotypic variation [29].

According to our current understanding, there is little data on the immunological changes and genetic variations linked to inflammatory postpartum disorders in buffalo, and there are also few reliable diagnostic and prognostic tools to help us improve our preventative and therapeutic measures. Furthermore, it is important to emphasize that more research is necessary to identify any relevant immunological indicators associated with reproductive disorders that determine whether buffalo are resistant to or susceptible to these diseases. In this study, real-time PCR and PCR-DNA sequencing techniques were used to assess the efficiency of putative immune genes as candidates for inflammatory reproductive disease incidence prediction and tracking in Italian buffaloes.

## 2. Material and Methods

### 2.1. Dairy Buffaloes and Research Samples

The 120 Italian buffaloes used in this study were grown on a private farm in the Delta region of northern Egypt, 60 of which had inflammatory reproductive problems, and the remaining 60 appeared to be healthy. The investigated female buffaloes were aged 4.5 years and in the second lactation season with an average milk yield of 15 ± 2.7 kg. Researched animals were grown in a commercial dairy herd with approximately 300 dairy buffaloes. The healthy group had a history of regular reproductive performance (i.e., normal feed intake, body temperature, no uterine discharge, and normal udder). The buffaloes in the second group exhibit mastitis (high body temperature, low appetite, swollen and tender udder, reddish and yellowish milk color and bad odor, clotted milk, and teat cracks) and endometritis (pyrexia, persistent colored uterine discharge with offensive odor, anorexia, and depression). Each buffalo’s jugular vein was pierced to get five milliliters of blood. The samples were placed into tubes encompassing anticoagulants (EDTA or sodium fluoride) in order to collect entire blood and extract DNA and RNA. All animal management techniques, test experimental collection and sample eradication were carried out under the direction of the Veterinary Medical School of the University of Sadat City (Code VUSC-027-1-23) in accordance with IACUC guidelines.

### 2.2. DNA Extraction and Amplification

The genetic material JET complete blood genomic DNA isolation kit and the manufacturer’s instructions (Thermo Scientific, Vilnius, Lithuania) were used to extract DNA from the genome using total blood. DNA with good purity and concentration was taken into consideration using Nanodrop. PCR was conducted on a conserved part, i.e., exon of investigated genes that enables accurate molecular characterization of genes and deciphers physiological differences in resistance/susceptibility to inflammatory postpartum disorders. Therefore, parts of coding sites (CDS) for immunity-related genes (*IKBKG*, *LGALS*, *IL1B*, *CCL2*, *RANTES*, *MASP2*, *HMGB1*, and *S-LZ*) were amplified. The *Bubalus bubalis* genome, which was accessible in GenBank, was utilized to create the oligonucleotide sequences for amplification. The primers used during the PCR are listed in Table 1.

A heat cycler with a final volume of 150 μL processed the polymerase chain cloning mixture. Every reaction container contained the following components: 66 μL d.d. water, 6 μL genetic material, 1.5 μL of each primer pair, and 75 μL of the master combination (Jena Bioscience, Jena, Germany). At a starting temperature of 94 °C for unwinding, the PCR combinations were used for five minutes. The 35 cycles consisted of 30-s rounds 30-s rounds of denaturation at 94 °C for 1 min, one-minute annealing cycles depending on the temperature range shown in Table 1, elongation at 72 °C for 1 min, and final elongation at 72 °C for 8 min. The materials were kept at 4 °C. Representative results of PCR analysis were detected by agarose gel electrophoresis. The fragment patterns were then visualized under U.V using a gel documentation system.

### 2.3. Discovering Polymorphisms

Preceding DNA sequencing, kits from Jena Bioscience # pp-201s/Munich of Hamburg, Germany, presented techniques to purify PCR in order to remove primer dimmers, non-specific bands, and other impurities and provide the desired cloned PCR product for the projected scope [30]. When assessing PCR output, a Nanodrop (Waltham, MA, USA, UV-Vis spectrometer Q5000) was used because it provided adequate quality and good concentrations [31]. Using normal and inflammatory reproductive disorders affected buffaloes, sequence analysis of the PCR-produced amplification results was used to identify SNPs. Using the enzyme chain terminator method outlined by Sanger et al. [32], the PCR products were sequenced on an ABI 3730XL DNA sequencer (United States: Applied Biosystems, Waltham, MA, USA).

Chromas 1.45 and BLAST 2.0 were the tools used for examining the DNA analysis findings [33]. Comparing the immune gene PCR results to the GenBank-provided reference gene sequences revealed polymorphisms. The MEGA6 tool can detect differences in the amino acid categorizations among the examined genes according to sequence matching among the buffaloes under investigation [34].

### 2.4. Levels of Immune Gene Transcripts

The RNA was completely isolated using the Trizol solution per the manufacturer’s instructions from the blood samples collected from the studied buffaloes (RNeasy Mini Ki, 74104, Product No., Qiagen, Venlo, The Netherlands). We quantified and confirmed the amount of RNA that was isolated using a NanoDrop^®^ ND-1000 spectrophotometer. The producer’s method was used to create the whole nucleic acid in each sample (Waltham, MA, USA: Thermo Fisher, Product No. EP0441). SYBR Green PCR Master Mix and quantitative RT-PCR (2× SensiFastTM SYBR, Bio-line, CAT No. Bio-98002) were used to assess the immune genes’ expression profiles. SYBR Green PCR Master Mix (Toronto, ON, Canada: Quantitect SYBR green PCR reagent, Catalogue No. 204141) has been used to determine the relative amount of mRNA that each sample possesses.

The sense and anti-sense primer oligonucleotides were made using the *Bubalus bubalis* genome assembly (NDDB_SH_1) from GenBank (Table 2). The constitutive normalization reference was the *ß. actin* gene The reaction mixture in an overall volume of 25 microliters consisted of entire RNA, 3 microliters, 4 microliters of 5xTrans Amp buffer, 0.25 microliters of reverse transcriptase, 0.5 microliters of each primer, 12.5 microliters of 2x Quantitect SYBR green PCR master mix, and 8.25 microliters of RNase-free water. Next, the prepared reaction mixture underwent the following procedures inside a heater cycler: reverse transcription for 30 min at 55 °C, preparatory denaturation for 8 min at 95 °C, 40 cycles of 15 s at 95 °C, and primer binding temperatures as listed in Table 2 with a 1-min extension at 72 °C. Following the amplifying process, a melting curve analysis was used to show the specificity of the amplified product. The 2^−ΔΔCt^ technique has been used to investigate the variations in each gene’s expression by comparing each gene’s expression in the examined sample to that of the *ß. actin* gene [35,36].

### 2.5. Statistical Analysis

H_o_: The resistance/susceptibility of inflammatory reproductive diseases in Italian buffaloes could not be explained by polymorphisms and transcript levels of immunological biomarkers.

H_A_: The resistance/susceptibility of inflammatory reproductive diseases in Italian buffaloes could be explained by polymorphisms and transcript levels of immunological biomarkers.

Using Fisher’s exact test analysis (*p* < 0.01), the significant distribution of SNPs for the identified genes was observed between the buffaloes under study. The *t*-test was used to determine the statistical significance of the differences in the expression profile of immune genes between normal and inflammatory reproductive illness-affected buffaloes. Mean and standard error (mean ± SE) were used to present the results. Differences were deemed significant at *p* < 0.05. 

## 3. Results

### 3.1. Immune Gene Genetic Polymorphisms

SNP variations in amplified DNA nucleotides associated with inflammatory reproductive disorders were found in the results of PCR-DNA sequencing on both normal and affected buffaloes for the IKBKG (412-bp), LGALS (395-bp), IL1B (358-bp), CCL2 (347-bp), RANTES (399-bp), MASP2 (513-bp), HMGB1 (540-bp), and S-LZ (455-bp) genes. The DNA sequence differences between the GenBank-sourced reference gene nucleotide sequence and the reproductive indicators examined in the studied buffaloes were utilized to verify all of the discovered SNPs (Appendix A). Table 3 shows the dissemination of a single base variation as well as a type of inherited change for immune indicators in normal and inflammatory reproductive disorder-affected buffaloes. The SNP Fisher’s exact test analysis revealed considerably different occurrences of the investigated markers in the normal and inflammatory reproductive disorder-affected buffaloes (*p* < 0.01). All of the reproductive markers under research had exonic region mutations, which resulted in different coding DNA sequences in normal buffaloes versus inflammatory reproductive disorder-affected ones. The immunological indicators that were being examined all had the exonic region modifications shown in Table 3 that led to coding mutations between the inflammatory reproductive disorder buffaloes and the healthy ones. Eighteen SNPs were discovered using DNA sequencing of immune genes; 10 of them are non-synonymous, and 8 are synonymous. 

The *IKBKG* gene (412-bp) contained two non-synonymous SNPs, C78G and G328A, which caused the amino acids T26M and G110S to be substituted, respectively. For the LGALS gene (395-bp), synonymous mutation 64N occurred as a result of T192C SNP. The nucleotide sequence of the IL1B gene (359-bp) revealed four recurrent non-synonymous SNPs: T18M, V42A, L69S, and S104L, resulting from C53T, T125C, T205C, and C311T SNPs, respectively. One recurrent SNP in the CCL2 gene (347-bp) was discovered using DNA sequencing; C201T resulted in synonymous mutation, 67P. For the RANTES gene (399-bp), three observed recurrent synonymous SNPs, G96C, A141G, and C370T, resulted in 32T, 47S, and 124L amino acids, respectively. One recurrent SNP was found when the MASP2 gene (343-bp) was sequenced, where synonymous mutation 155A occurred as a result of T465A SNP. Three recurrent SNPs in the HMGB1 gene (540-bp) were discovered using DNA sequencing: C227A, C275T, and T407C, which resulted in non-synonymous mutations, S76L, P92L, and F136S, respectively. Three recurrent SNPs in the S-LZ gene (455-bp) were discovered using DNA sequencing; two of these, G33A and A222G, resulted in synonymous mutations, 11R and 74L, respectively. In comparison, the non-synonymous mutation caused by the G254A SNP led to the substitution of the amino acid R85K.

### 3.2. Immune Markers’ Transcriptional Levels Tendencies

In inflammatory reproductive disorder-affected buffaloes, expression levels of the genes *IKBKG*, *LGALS*, *IL1B*, *CCL2*, *RANTES*, *MASP2*, *HMGB1*, and *S-LZ* were noticeably upregulated (Figure 1). The highest possible level of mRNA for each gene determined in the buffaloes with the inflammatory reproductive diseases was found for *CCL2* (2.21 ± 0.18), and the lowest level was found for *LGALS* (1.47 ± 0.15). Among all the genes analyzed in the normal buffaloes, the gene with the greatest possible mRNA level was found for *S-LZ* (0.65 ± 0.06), while the lowermost level was found for *IKBKG* (0.46 ± 0.15).

## 4. Discussion

### 4.1. Association of Immune Genes Polymorphisms with Inflammatory Reproductive Disorders

Knowing the genes, fundamental mutations, and communications with other variables that confer resistant animals is crucial for the efficient exploitation of disease-resistant cattle or the complete elimination of diseased animals [37]. The immune (*IKBKG*, *LGALS*, *IL1B*, *CCL2*, *RANTES*, *MASP2*, *HMGB1*, and *S-LZ*) genes in inflammatory reproductive disorder-affected and healthy Italian buffaloes were identified in this study utilizing sequenced amplified PCR products. The findings show that there are differences between the SNPs involving the healthy and inflammatory reproductive disorder-affected buffaloes. The assessed buffaloes had a considerable nucleotide polymorphism dispersion (*p* < 0.01), according to the Fisher’s exact test. It is important to emphasize that the polymorphisms found and made available in this context give new data for the evaluated indicators when compared to the pertinent datasets collected from GenBank.

Our findings demonstrated the presence of two nonsynonymous SNPs, C78G and G328A, in the *IKBKG* gene (412-bp). According to the results of the basal local alignment search tool (BLAST), the cattle database’s altered mutated bases are conserved (GenBank accession number XM_027535016). Additionally, databases for sheep and goats both have the G328A SNP (GenBank accession numbers XM_042241635.1 and XM_013976823.2, respectively). A synonymous mutation for the *LGALS* gene (395-bp) was brought on by the T192C SNP. Cattle (GenBank accession number XM_027540847.1) retain the altered base. Four recurring non-synonymous SNPs, C53T, T125C, T205C, and C311T SNPs, were identified in the 359-bp nucleotide sequence of the *IL1B* gene. The identified SNPs were found to be conserved in other closely related species by nucleotide sequence alignment. For example, the goat has a conserved C53T SNP (GenBank accession number DQ837160.1), the cattle have a conserved T205C SNP (GenBank accession number EU276067.1), and the *Syncerus caffer* (African buffalo) has a conserved C311T mutated base (GenBank accession number AB571651.1). DNA sequencing led to the discovery of one recurrent SNP (in the *CCL2* gene 347-bp): C201T. When compared to the genome of cattle, the changed nucleotide was shown to be conserved (GenBank accession number EU276059.1).

Similarly, when comparing our DNA sequencing results with those of closely related species, three recurrent synonymous SNPs, G96C, A141G, and C370T, were found in the *RANTES* gene (399 bp). In the genome of cattle, the three altered nucleotides are conserved (GenBank accession numbers CP027087.1, XM_027518916.1, and OX344708.1). The A141G SNP was also discovered to be conserved in databases for sheep and goats, with the GenBank accession numbers XM_027975305.2 and XM_005693201.3, respectively. The recurrent T465A SNP for the *MASP2* gene (343-bp) sequence was reported in nucleotide sequences from sheep and cattle, respectively, with the GenBank accession codes CP011908.1 and CP027092.1. C227A, C275T, and T407C are three recurrent SNPs that were found using DNA sequencing in the 540-bp *HMGB1* gene. By comparing the three detected SNPs to cattle and sheep DNA sequences stored in GenBank under accession numbers CP027087.1, XM_042254827.1, CP011895.1, and LR962873.1, it was discovered that they were identical. However, only the C227A and T407C SNPs (GenBank accession numbers XM_018056617.1 and XM_018056617.1) were linked to the goat sequence. DNA sequencing was used to identify three recurrent SNPs in the S-LZ gene (455 bp), two of which, G33A and A222G, led to synonymous alterations. While the G254A SNP-caused non-synonymous mutation. The sheep GenBank sequence with accession numbers AH008120.2 and CP011908.1 matched the three newly identified SNPs. While the goat sequence shared the identical changed base for the G254A SNP (GenBank under accession number KC954666.1), the G33A was discovered to be aligned with the cattle database with the GeBank number M95099.1. It should be noted that the A222G SNP has the identical altered base in both cattle and goats, with accession numbers that may be found in GenBank: MF996700.1 and KC526216.1, respectively. The conservation behavior in the altered bases may be attributed to close relatedness among ruminant species, where genetic resource conservation programs are contributing to an increase in the numbers and to the preservation of valuable gene reservoirs [38]. Another cause is conducting PCR-DNA sequencing on the conserved part (CDS) of the investigated immune genes [39]. Little information is available on polymorphisms and expression profiles of immune biomarkers in inflammatory postparturient disorders in buffaloes. The initial evidence for this connection can be found in the gene sequences from the *Bubalus bubalis* that were used in our work and were published in PubMed. The variance of the immune (*IKBKG*, *LGALS*, *IL1B*, *CCL2, RANTES*, *MASP2*, *HMGB1*, and *S-LZ*) markers and how they relate to inflammatory reproductive disorders in Italian buffaloes have not, to our knowledge, been the subject of any prior research. The candidate gene method has, however, been utilized to assess the veracity of postpartum problems in buffalo. For instance, sequence analysis variations of the *TLR2* and *TLR4* genes were linked to endometritis and mastitis in river buffalo [40,41,42]. 

In order to better understand the resistance and vulnerability of inflammatory reproductive diseases in livestock, genetic polymorphisms of immunological markers for postpartum endometritis were examined in Holstein dairy cows [43]. Using PCR-DNA sequencing for the immunological (*TLR4*, *IL10*, *TLR7*, *NCF4*, *TNF-α*, and *LITAF*) genes, there were changes in the nucleotide sequence between endometritis-affected cows and healthy ones. In addition, using PCR-DNA sequencing of immunological (*SELL*, *ABCG2*, *FEZL*, and *SLC11A1*) markers from the investigated Holstein and Brown Swiss dairy cows, nucleotide sequence alterations in the form of SNPs related to mastitis tolerance/susceptibility were found [44]. In cows, the *LTF* gene was linked to mastitis resistance [45,46,47]. *TLR4* gene polymorphisms and mastitis susceptibility were found to significantly correlate in sheep [48] and cattle [49,50]. According to Ruiz-Rodriguez et al. [51], there is a correlation between goats’ susceptibility to or resistance to mastitis and variations in the *TLR2* gene. In Barki sheep, *TLR4* gene SNPs were likewise linked to postpartum disorder risk [27].

The primary source of adaptation and selection is mutation [52]. All of the immune markers that were being studied in this situation had exonic region mutations, which led to altered coding DNA sequences in repeat breeder buffaloes compared to healthy ones. Using DNA sequencing of immune genes, eighteen SNPs were found; 8 of them are synonymous, and 10 are not. Non-synonymous mutations alter protein sequences, and animals that have these mutations are frequently the subject of natural selection [52]. The encoded amino acid at the mutant location is altered by genetic variation brought on by non-synonymous SNPs, which can result in structural and functional alterations in the mutated protein [53]. For a very long time, it was believed that selection on synonymous mutations was either nonexistent or very weak [52]. In order to accurately characterize the investigated immune genes at the molecular level and to understand the physiological differences in resistance/susceptibility between normal and inflammatory reproductive disorders buffaloes, our study discovered polymorphisms on the basis of translated DNA sequence to be of greater value than intronic parts.

### 4.2. Immune Markers’ Gene Expression Profile

In the current work, we proposed that the course of inflammatory reproductive disorders may be influenced by genetic variation in an individual’s transcriptional response to those conditions. *IKBKG*, *LGALS*, *IL1B*, *CCL2*, *RANTES*, *MASP2*, *HMGB1*, and *S-LZ* genes were measured using real-time PCR in both normal and inflammatory reproductive disorder-affected buffaloes. Our results demonstrated that buffaloes with inflammatory reproductive diseases expressed immune genes at higher levels than healthy ones. In order to ascertain the mRNA levels of these immunological markers linked to the resistance/susceptibility to inflammatory reproductive diseases in Italian buffaloes, real-time PCR was used in our experiment for the first time. Our investigation utilized qualitative SNP genetic markers, and quantitative gene expression overcame the drawbacks of past studies. As a result, in both healthy and affected by inflammatory reproductive illness buffaloes, the analyzed gene regulation’s mechanisms have been well-established.

Buffaloes suffering from reproductive problems have a sparsely developed gene expression profile of immunological markers. The findings of a study on the expression of cytokine genes in the uteri of *Bubalus bubalis* associated with endometritis infection revealed that *IL-1* and *IL-6* gene expression was upregulated by 1.3, 1.7, and 5-fold, respectively, in endometritis-infected buffaloes compared to control animals. In contrast, when compared to uninfected animals, the expressions of *IL-10* and *TNF-α* revealed lowered mRNA of 0.2 and 0.4-fold, respectively [54].

In dairy cows, the bovine endometrium of cows with inflammation had greater transcript levels of immunological (*CCL5*, *CXCL8*, *IL6*, and *IL1B*) genes than cows without inflammation [55]. In mastitic Chinese Holstein cattle, *TLR4*, *MyD88*, *IL-6*, and *IL-10* were upregulated at the transcript level, while *CD14*, *TNF-*, *MD-2, NF-B*, and *IL-12* were noticeably downregulated [56]. When compared to resistant cows, endometritis-affected cows had considerably higher levels of the genes *TLR4*, *TLR7*, *TNF*, *NCF4*, and *LITAF* expression. Meanwhile, the expression of the *IL10* gene was much decreased [43]. *SELL*, *SLC11A1*, and *FEZL* gene expression were significantly higher in mastitic Holstein and Brown Swiss dairy cows than in tolerant animals [46]. Specific mRNA expression patterns of *C3*, *C2*, *LTF*, *PF4*, and *TRAPPC13* are present in the blood and endometrium of dairy calves with subclinical endometritis. Additionally, in contrast to cows with sub-clinical endometritis, the gene expression profiles of *C3*, *CXCL8*, *LTF*, *TLR2*, and *TRAPPC13* were temporally changed in healthy cows’ circulating WBC during the postpartum period [57]. *CCL5* gene expression did not significantly alter between the three different time periods during the transition phase [18]. In goats, the *LTF* gene for mastitis resistance was discovered to have an up-regulated expression profile in Damascus goats [58]. Goats with *Staphylococcus aureus* mastitis were found to have a considerable increase in pro-inflammatory cytokines, chemokines, and their receptors compared to tolerant goats [59]. In the goat model of subclinical endometritis created by Shao et al. [60], *TLR4*, cytokines, and beta-defensin dramatically increased in mRNA expression. Regarding the sheep’s gene expression profile for immunological markers, according to Darwish et al. [27], sheep with postpartum problems had considerably higher mRNA of *IL5*, *IL6*, *IL1-ß*, *TNF alpha*, *TLR4*, and *Tollip* than tolerant ewes. 

The inhibitor of nuclear factor kappa B kinase regulatory subunit gamma (*IKBKG*) gene encodes the regulatory subunit of the inhibitor of kappaB kinase (IKK) complex. When NF-kappaB is activated, genes implicated in cell survival and immunity are also activated [61]. Animal lectins known as galectins have an affinity for β-galactosides [62]. Through lectin-carbohydrate interactions, they can interact with glycoproteins on the surface of cells and in the extracellular matrix [62]. Galectins influence cell survival, alter cell adhesion, and trigger cell migration through this process. Numerous cell and tissue types contain them, and a diversity of their functions, such as encouraging macrophage migration and playing a part in both acute and chronic inflammation, have been documented [63]. In inflammatory circumstances, cytokines such as IL-6, TNF-α, IFN-γ, IL1B, and NFKB serve as indirect indicators [64]. 

C-C Motif Chemokine Ligand 2 (*CCL2*), which encodes two tiny proteins that bind to the CCR2 receptors present on the surface of monocytes and neutrophils, respectively, acts as a potent chemokine for these cells [65]. These molecules interact with receptors on neutrophils and monocytes to cause conformational changes, which permit cytosolic free Ca^2+^ to encourage chemotaxis towards inflammatory stimuli [66]. The relationship between single nucleotide polymorphisms in the bovine *CCL2* gene and production and health was investigated using Canadian Holstein cattle [67]. Numerous cells, including blood lymphocytes, express the chemokine regulated on activation normal T-cell expressed and secreted (RANTES), also known as CCL5, in response to inflammatory signals [68]. It controls both inflammatory and non-inflammatory cells’ migration and activation [69]. It also has an impact on the acute phase response [70].

Mannose-binding lectin-associated serine protease 2 (MASP2) serves as the primary protease in the complement system [71]. Autoimmune disorders possess a substantial association with polymorphisms in the *MASP2* gene [72]. Numerous inflammatory illnesses and infections have been associated with MASP2 serum levels and gene variants [73]. It was shown how the polymorphisms of the *MASP2* gene relate to mastitis and milk output in Chinese Holstein cattle [74]. High-mobility group box 1 (HMGB1) is a protein with important biological roles that is both abundant and conserved. It can stimulate the CXCR4 to increase chemotaxis by binding to the chemokine CXCL12 or activate the toll-like receptor 4 (TLR4) to generate cytokines [75]. In addition to its intrinsic action, HMGB1 interacts with PAMPs, cytokines, and chemokines to extend its extracellular effects [76]. An anti-bacterial peptide called lysozyme is essential for giving animals immunity [77]. Lysozyme plays a substantial role in treating cow mastitis, according to earlier investigations [77,78].

Italian buffaloes with inflammatory reproductive disorders exhibit a marked upregulation in the expression of immune (*IKBKG*, *LGALS*, *IL1B*, *CCL2*, *RANTES*, *MASP2*, *HMGB1*, and *S-LZ*) markers. This could be explained by cytotoxic radicals and proinflammatory cytokines generated by the phagocytic cells, both of which cause serious inflammation that harms the damaged tissue [79,80]. Additionally, the immune system is compromised by the preponderance of ROS caused by the excess ROS in the absence of an optimum entire antioxidant [80]. Inflammation of the reproductive tract is frequently brought on by bacterial infections [81]. The buffaloes’ immune systems are triggered as a result of being exposed to more infections. When macrophages and epithelial cells are exposed to either lipoteichoic acid (LTA, from Gram-positive bacteria) or LPS, from Gram-negative bacteria, which both induce the release of TNF and IL1B, the neutrophil recruitment cascade is started [82]. Vascular endothelial cells respond when TNF, IL1, C5a, and histamine engage with their specific receptors [83]. The vast majority of the inflammatory reproductive disease cases in our study that impacted buffaloes are, therefore, thought to have been caused by an infectious agent. Furthermore, the results of our real-time PCR analysis provide strong proof that the inflammatory disorders that plagued the buffaloes were causing them to experience a considerable inflammatory response. Our idea is that variations in the essential genes associated with these defense systems can lead to variations in disease resistance. Buffaloes have essential, shared innate immune defense mechanisms against multiple reproductive tract infection sources. 

## 5. Conclusions

Single nucleotide variations (SNPs) in the genes for immunological (*IKBKG*, *LGALS*, *IL1B*, *CCL2*, *RANTES*, *MASP2*, *HMGB1,* and *S-LZ*) indicators were discovered using PCR-DNA sequencing in healthy and sick Italian buffaloes with inflammatory reproductive diseases. Furthermore, healthy and affected buffaloes with inflammatory reproductive disorders have different levels of these indicators’ mRNA. By employing genetic markers along with normal welfare during buffalo selection, a promising opportunity exists to lessen the occurrence of inflammatory reproductive disorders thanks to these novel functional variations. The gene domains identified here may facilitate future methods for treating inflammatory reproductive diseases.

## Figures and Tables

**Figure 1 vetsci-10-00573-f001:**
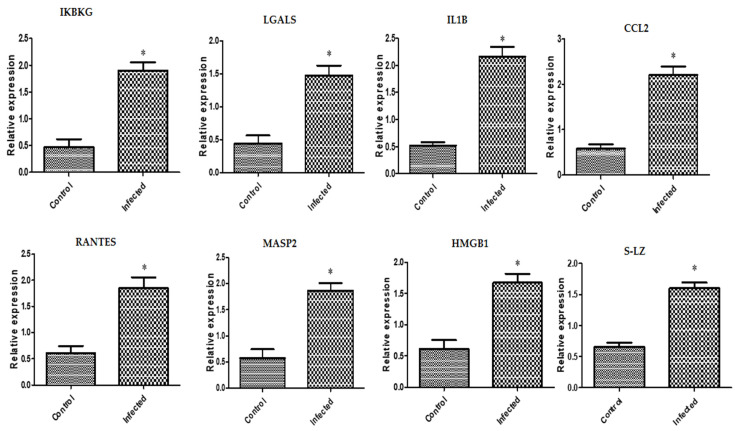
Different immune gene transcript levels between normal and inflammatory reproductive disorders affected buffaloes. The symbol * denotes significance when *p* < 0.05.

**Table 1 vetsci-10-00573-t001:** Immune gene oligonucleotide primers for the investigation of genetic polymorphisms.

Investigated Gene	Sense	Antisense	Accession Number	Annealing Temperature (°C)	Size of PCR Product (bp)
*IKBKG*	5′-CCGGCTTGGACAGCCTCTCTTG-3′	5′-TGCCCTGCTCTGAAGGCACATG-3′	XM_044936650.2	58	412
*LGALS*	5′-CTCATTTCTCCTGCAGAGTCT-3′	5′-AACTTGAATTCGTATCCATCAG-3′	XM_025282898.3	58	395
*IL1B*	5′-CCATGGCAACCGTACCTGAAC-3′	5′- ACTGACTGCACGGGTGCGTCAC-3′	NM_001290898.1	58	359
*CCL2*	5′-CTTGAATCCTCTCGCTGCAAC-3′	5′-AGATTCTTGGGTTGTGGAGTGA-3′	HQ889748.1	60	347
*RANTES*	5′-CAGCCGGAGCTGCAGAGGATCA-3′	5′- CAAGCTGCTTAGGACAAGAGCG-3′	XM_006056212.4	58	399
*MASP2*	5′-AGAGCACAGACACAGAGCGGGC-3′	5′- AGATGCGGCCATTAGGTGGCGC-3′	XR_006551148.1	60	513
*HMGB1*	5′-CGCTGGCTGGAGAGTAATGTTA-3′	5′-CACCAATAGACAGGCCAGGATGT-3′	XM_006078587.2	60	540
*S-LZ*	5′-TCTGGACATTTGACTTCTCAG-3′	5′-CTCAATGTAACTGCTGACGTC-3′	EF535848.1	58	455

*IKBKG =* Inhibitor of nuclear factor kappa B kinase regulatory subunit gamma; *LGALS =* Galectin; *IL1B* = Interleukin 1 beta; *CCL2* = C-C Motif Chemokine Ligand 2; *RANTES* = Regulated on activation, normal T cell expressed and secreted; *MASP2* = Mannan-binding lectin serine protease-2; *HMGB1* = High mobility group box 1; and *S-LZ* = serum lysozyme.

**Table 2 vetsci-10-00573-t002:** Oligonucleotide-based real-time PCR primers for the under investigation immunological genes.

Investigated Gene	Primer	Product Size (bp)	Annealing Temperature (°C)	GenBank Isolate	Origin
*IKBKG*	F5′-CTGGCAGGAGAAGCCATCAA-3 R5′-CAGCAGACAGGACACTAGCC-3′	101	58	XM_044936650.2	Current study
*LGALS*	F5′-GGCCCAAAGCTCATTTCTCC-3 R5′-GCACTCCCCAGGTTTGAGAT-3′	129	58	XM_025282898.3	Current study
*IL1B*	F5′-AGGTGGTGTCGGTCATTGTG-3′ R5′-AACTCGTCGGAGGACGTTTC-3′	142	60	NM_001290898.1	Current study
*CCL2*	F5′-AAGCCTTGAGCACTCACTCC -3′ R5′-GCAGTTAGGGAAAGCCGGAA-3′	72	56	HQ889748.1	Current study
*RANTES*	F5′-CCCATATGCCTCAGACACCAC-3′ R5′-GGCGGTTCTTCCTGGTGATA-3′	140	60	XM_006056212.4	Current study
*MASP2*	F5′-ACAGACAAGGCGGAATACGG-3′ R5′-GTGGTCTACAGCAGGCAAGT-3′	162	58	XR_006551148.1	Current study
*HMGB1*	F5′-TGCCTCGCGGAGGAAAAATA-3′ R5′-GCAGACATGGTCTTCCACCT-3′	188	60	XM_006078587.2	Current study
*S-LZ*	F5′-CTGTAGCCTGTGCAAAGCAG -3′ R5′-CAGGGTGCAACCCTCAATGT-3′	113	60	EF535848.1	Current study
*ß. actin*	F5′-GATGATGATATTGCCGCGCTC-3′ R5′-AGGGTCAGGATGCCTCTCTT-3′	197	58	NM_001290932.1	Current study

*IKBKG =* Inhibitor of nuclear factor kappa B kinase regulatory subunit gamma; *LGALS =* Galectin; *IL1B* = Interleukin 1 beta; *CCL2* = C-C Motif Chemokine Ligand 2; *RANTES* = Regulated on activation, normal T cell expressed and secreted; *MASP2* = Mannan-binding lectin serine protease-2; *HMGB1* = High mobility group box 1; and *S-LZ* = serum lysozyme.

**Table 3 vetsci-10-00573-t003:** Single base difference dispersal, as well as the sort of inherited change for immune markers in healthy and inflammatory reproductive disorder-affected buffaloes.

Gene	SNPs	Healthy *n* = 60	Reproductive Disorders *n* = 60	Total *n* = 120	Sort of Inherited Change	Amino Acid Order and Kind
*IKBKG*	C78G	0/60	31/60	31/120	Non-synonymous	26 I to M
	G328A	27/60	0/60	27/120	Non-synonymous	110 G to S
*LGALS*	T192C	42/60	0/60	42/120	Synonymous	64 N
*IL1B*	C53T	19/60	0/60	19/120	Non-synonymous	18 T to M
	T125C	47/60	0/60	47/120	Non-synonymous	42 V to A
	T205C	0/60	23/60	23/120	Non-synonymous	69 L to S
	C311T	49/60	0/60	49/120	Non-synonymous	104 S to L
*CCL2*	C201T	0/60	53/60	53/120	Synonymous	67 P
*RANTES*	G96C	0/60	29/60	29/120	Synonymous	32 T
	A141G	0/60	43/60	43/120	Synonymous	47 S
	C370T	54/60	0/60	54/120	Synonymous	124 L
*MASP2*	T465A	22/60	0/60	22/120	Synonymous	155 A
*HMGB1*	C227T	34/60	0/60	34/120	Non-synonymous	76 S to L
	C275T	0/60	29/60	29/120	Non-synonymous	92 P to L
	T407C	56/60	0/60	56/120	Non-synonymous	136 F to S
*S-LZ*	G33A	39/60	0/60	39/120	Synonymous	11 R
	A222G	0/60	52/60	52/120	Synonymous	74 L
	G254A	0/60	37/60	37/120	Non-synonymous	85 R to K

Single base difference dispersal for immune markers in healthy and reproductive disorders buffaloes showed a highly significant variation (*p* < 0.01) according to Fisher’s exact analysis. *IKBKG =* Inhibitor of nuclear factor kappa B kinase regulatory subunit gamma; *LGALS =* Galectin; *IL1B* = Interleukin 1 beta; *CCL2* = C-C Motif Chemokine Ligand 2; *RANTES* = Regulated on activation, normal T cell expressed and secreted; *MASP2* = Mannan-binding lectin serine protease-2; *HMGB1* = High mobility group box 1; and *S-LZ* = serum lysozyme. A = Alanine; F = Phenylalanine; G = Glycine; I = Isoleucine; K = Lysine; L = Leucine; M = Methionine; N = Asparagine; P = Proline; R = Argnine; S = Serine; T = Threonine; and V = Valine.

## Data Availability

Upon justifiable demand, Appendix A for the study’s conclusions will be given by the corresponding author.

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
