# Peer review of "DNA Polymorphisms and mRNA Levels of Immune Biomarkers as Candidates for Inflammatory Postpartum Disorders Susceptibility in Italian Buffaloes"

_vetsci, 2023, doi:10.3390/vetsci10090573_

Round 1
Reviewer 1 Report
Comments to the Author:
Postnatal inflammatory diseases of buffalo reduce conception rate and age of buffalo and have a detrimental effect on the economic yield of herd farming. In this manuscript, the authors used PCR-DNA sequencing to identify differences in the nucleotide sequences of immune genes between healthy buffalo and those with inflammatory reproductive disorders. When buffalo experienced inflammatory reproductive disorders, they were significantly more likely to express the genes examined. Therefore, these molecular changes may provide important information on the physiological relationships of various reproductive pathways. The author's analysis is very detailed. However, some issues need to be resolved before they are received.
Point 1: Each line should be labeled during the manuscript review process for ease of access, and this manuscript is not labeled with line numbers.
Point 2: The content of the “2.1 Dairy Buffaloes and Research Samples” part of the manuscript is more verbose and requires a more concise description. (For example, the authors described twice that the buffaloes used in the study could be divided into healthy and those with inflammatory reproductive problems, it is recommended to rewrite it and describe it only once to make it look more streamlined.)
Point 3: In the sentence “and recover DNA and RNA”, would it be better to replace the “recover” with a “extract”.
Point 4: The “the two categories” described in the “4.1 Association of Immune Genes Polymorphisms with Inflammatory Reproductive Disorders” section needs to specifically describe those two categories.
Point 5: In part “4.2. Immune Markers’ Gene Expression Profile” of the manuscript, the author proposed “using SNP genetic markers and gene expression to study gene polymorphism”, but in the whole manuscript, it only showed that the expression of these genes was higher in buffaloes with inflammatory reproductive disorders, which had no obvious relationship with polymorphism.
Point 6: The format of manuscripts and references needs to be revised in detail according to the requirements of the journal.
-
The entire manuscript needs to be examined and slightly revised.
Author Response
Reviewer 1
Comments and Suggestions for Authors
Comments to the Author:
Comment
Postnatal inflammatory diseases of buffalo reduce conception rate and age of buffalo and have a detrimental effect on the economic yield of herd farming. In this manuscript, the authors used PCR-DNA sequencing to identify differences in the nucleotide sequences of immune genes between healthy buffalo and those with inflammatory reproductive disorders. When buffalo experienced inflammatory reproductive disorders, they were significantly more likely to express the genes examined. Therefore, these molecular changes may provide important information on the physiological relationships of various reproductive pathways. The author's analysis is very detailed. However, some issues need to be resolved before they are received.
Response
We thank reviewer for this positive comment.
Comment
Point 1: Each line should be labeled during the manuscript review process for ease of access, and this manuscript is not labeled with line numbers.
Response
We are grateful to the reviewer for drawing it to our consideration. The manuscript is labeled with line numbers.
Comment
Point 2: The content of the “2.1 Dairy Buffaloes and Research Samples” part of the manuscript is more verbose and requires a more concise description. (For example, the authors described twice that the buffaloes used in the study could be divided into healthy and those with inflammatory reproductive problems, it is recommended to rewrite it and describe it only once to make it look more streamlined.)
Response
We are grateful to the reviewer for drawing it to our consideration. The sentence is rewritten and described once to make it look more streamlined.
Comment
Point 3: In the sentence “and recover DNA and RNA”, would it be better to replace the “recover” with a “extract”.
Response
We thank reviewer for this. The word recover is replaced by extract.
Comment
Point 4: The “the two categories” described in the “4.1 Association of Immune Genes Polymorphisms with Inflammatory Reproductive Disorders” section needs to specifically describe those two categories.
Response
We are grateful to the reviewer for drawing it to our consideration. The two categories are specifically described.
Comment
Point 5: In part “4.2. Immune Markers’ Gene Expression Profile” of the manuscript, the author proposed “using SNP genetic markers and gene expression to study gene polymorphism”, but in the whole manuscript, it only showed that the expression of these genes was higher in buffaloes with inflammatory reproductive disorders, which had no obvious relationship with polymorphism.
Response
- We thank reviewer for this. Concerning this sentence we mean that previous studies explored genetic polymorphisms using qualitative genetic markers (RFLP and DNA sequencing).
- Our study utilized qualitative SNP genetic markers and quantitative gene expression to overcome the drawbacks of past studies. This is illustrated in the manuscript by adding the words qualitative and quantitative.
- SNP as a genetic marker gives qualitative results (gene shape i.e. normal or mutated); however real time PCR is a quantitative (amount of mRNA that indicates amount of protein). In our investigation we combined the two approaches. As a result, in both healthy and affected by inflammatory reproductive illnesses buffaloes, the mechanisms of the examined gene regulation are well understood. Giving different results by real time PCR make the role of each studied marker in the susceptibility of economic diseases more obvious. There is not a guarantee that the two approaches give different results in the examined animals.
- Some previous studies reported that DNA sequencing of some candidate genes gives monomorphic pattern between examined animals i.e. there are no single nucleotide polymorphisms (SNPs) between examined animals. This means the investigated genes is not a candidate gene for the trait of interest. The selected gene may give a monomorphic pattern in the DNA sequence; however, therefore, it could give different results using quantitative real time. We determined to use qualitative and quantitative approaches in our investigation.
Comment
Point 6: The format of manuscripts and references needs to be revised in detail according to the requirements of the journal.
Response
We are grateful to the reviewer for drawing it to our consideration. The format of manuscript and references is revised in detail according to the requirements of the journal.
Comment
Comments on the Quality of English Language
- The entire manuscript needs to be examined and slightly revised.
Response
We are grateful to the reviewer for drawing it to our consideration. The entire manuscript is examined and slightly revised.

Reviewer 2 Report
Kind Authors,
I have read your work with interest and attention and, while recognizing the validity of your work, I believe that a number of changes must be made before any publication.
You will find my instructions in the attached file.

Author Response
Dear reviewer
Thank you very much for your interest and consideration. It is of great pleasure to receive your decision on our paper "DNA Polymorphisms and mRNA Levels of Immune Biomarkers as Candidates for Inflammatory Postpartum Disorders Susceptibility in Italian Buffaloes"
Please find our revised manuscript and the response to the reviewer comments. All comments were considered while revising this paper.

Round 2
Reviewer 2 Report
Dear authors,
I have carefully read the revised version of your article and I am satisfied with the changes made.
I just suggest you add the genome accession number to line 168